# Psychological Effects of Home Confinement and Social Distancing Derived from COVID-19 in the General Population—A Systematic Review

**DOI:** 10.3390/ijerph18126528

**Published:** 2021-06-17

**Authors:** Paula Rodríguez-Fernández, Josefa González-Santos, Mirian Santamaría-Peláez, Raúl Soto-Cámara, Esteban Sánchez-González, Jerónimo J. González-Bernal

**Affiliations:** 1Department of Health Sciences, University of Burgos, 09001 Burgos, Spain; prf0011@alu.ubu.es (P.R.-F.); rscamara@ubu.es (R.S.-C.); jejavier@ubu.es (J.J.G.-B.); 2Department of Health Sciences, University of Jan Kochanowski, 25-369 Kielce, Poland; estebansg2001@gmail.com

**Keywords:** COVID-19, pandemic, home confinement, social distancing, mental health, adults, general population, anxiety, depression, stress

## Abstract

(1) Background: Home confinement and social distancing are two of the main public health measures to curb the spread of SARS-Cov-2, which can have harmful consequences on people’s mental health. This systematic review aims to identify the best available scientific evidence on the impact that home confinement and social distancing, derived from the SARS-CoV-2 pandemic, have had on the mental health of the general population in terms of depression, stress and anxiety. (2) Methods: A systematic search was conducted on PubMed, Scopus, Web of Science and ScienceDirect between 2 January 2021 and 7 January 2021, in accordance with the recommendations of the PRISMA Declaration. The selection of studies and the evaluation of their methodological quality were performed in pairs, independently and blindly, based on predetermined eligibility criteria. (3) Results: The 26 investigations reviewed were developed in different regions and countries. Factors that are associated with poor mental health were female gender, young ages, having no income and suffering from a previous psychiatric illness. Inadequate management of the pandemic by authorities and a lack or excess of information also contributed to worse mental health. (4) Conclusions: There are groups of people more likely to suffer higher levels of anxiety, depression and stress during the restrictive measures derived from COVID-19.

## 1. Introduction

In December 2019, the SARS-CoV-2 emerged in Wuhan, China. The World Health Organization (WHO) declared the disease caused by this virus (the COVID-19) as the sixth international public health emergency and proclaimed the situation as a pandemic on the 11th of March of 2020 [1]. Since the pandemic began, many countries implemented public health measures, such as social distancing or home confinement, with the aim of minimizing the spread of this virus [2,3,4]. These interventions to protect the physical health of the population altered global patterns of behaviour producing changes in the economy, way of working, social interactions or daily life [5], which, in turn, can lead to an increase in the prevalence of health risk behaviours and psychological disorders [6,7,8,9].

Most studies that analyse the mental consequences in the general population of some previous epidemics and pandemics focus on symptoms related to the aftermath of the disease itself without taking into consideration the effects of social distancing or home confinement [10]. However, large-scale disasters have also been observed to increase the prevalence of different mental and behavioural disorders such as anxiety, depression, post-traumatic stress disorder (PTSD) or substance abuse [11,12,13,14]. We can take the Severe Acute Respiratory Syndrome (SARS) epidemic in 2002 as an example because this situation led to an increase in people with PTSD and psychological distress, and not only among people who suffered from the SARS disease but also among their relatives or health workers, symptoms that persisted for a long period of time [15]. Quarantine during the SARS outbreak was also associated with high rates of anxiety (28.9%) and depression (31.2%) [4].

The elderly and people with underlying diseases are particularly at risk for SARS-Cov-2 infection, but in terms of mental health as a result of measures to slow the spread of the virus, other factors appear to be contributing to the development of psychological symptoms during the pandemic [5,6,7,8,9]. For example, younger age has been linked to feelings of loneliness during the COVID-19 pandemic, leading to symptoms of anxiety and depression in this group [16]. In addition to economic losses, occupational deprivation and the pandemic itself, social isolation is the main cause of psychological symptoms during the COVID-19 [16]. Home confinement and social distancing during the pandemic have already been shown to be associated with adverse psychological outcomes, even in un-infected people [16,17], such as emotional disorders, depression, anxiety, stress, irritability, insomnia, PTSD, anger and emotional exhaustion [18], or risky behaviours and increased substance abuse [19]. This situation makes even more evident the need to pay attention to and strengthen the public’s mental health in order to minimize as much as possible the consequences of loneliness and social isolation due to the COVID-19. Therefore, investigation on this subject is justified to provide appropriate care, focused on the prevention and treatment of mental illnesses that will arise during and after the pandemic, as well as to establish programs and policies to support the global population during the crisis.

The main objective of this study was to identify the best available scientific evidence on the impact that home confinement and social distancing, derived from the SARS-CoV-2 pandemic, have had on the mental health of the general population in terms of depression, stress and anxiety.

## 2. Materials and Methods

In accordance with the recommendations of the PRISMA Declaration [20] and following the previously established research protocol, a systematic review of the scientific literature was made between 2 January and 7 January 2021. The electronic version of the following databases was consulted: *PubMed*, *Scopus*, *Web of Science* and *ScienceDirect*.

The search began with the formulation of a clinically answerable research question in PIO format, according to the criteria established by Sackett et al. [21] (Table 1).

Having asked that question, and according to it, different search strategies were designed and adapted to the particularities of each of the databases consulted. The appropriate “medical subjects headings” (MeSH), combined with boolean operators (AND/OR), together with free text terms, some of them truncated, were used in order to include all possible terminations (Table 2).

Those original research studies (1) with a cross-sectional or longitudinal descriptive type methodological design, (2) published in English, Spanish, French, Italian or Portuguese, (3) published from December 2019 onward, (4) with, at least, the summary being available, (5) that in their results evaluate the level of depression, stress and/or anxiety of the general population during the SARS-CoV-2 pandemic were selected. Clinical case reports, scientific letters or low-quality scientific records, and studies that did not answer the research question and were not related to the purpose of the review or those that analysed specific sub-groups of the population (children, youth, university students, health professionals, the elderly, people with specific diseases or pregnant women) were excluded.

As a secondary strategy, a manual reverse search (also known as “snowball searching”) was performed in order to identify possible relevant studies that were not previously taken into account. Sources of grey literature and bibliographic references cited in the selected studies were reviewed.

The selection of studies and the evaluation of their methodological quality were performed in pairs, independently and blindly, solving possible discrepancies by consensus and, if not, through the participation of a third evaluator. To ensure the homogeneity of all researchers in the collection of information, a standardized data extraction form was designed, including the following items for each of the selected articles: title and lead author, country and year of publication, type of study and objective, place and period of publication, sample size and characteristics, the definition of the analysed variables and instruments used, a brief summary of the obtained results and conclusions, along with the results of the evaluation of their scientific and technical quality. The “critical appraisal tools” of the Joanna Briggs Institute of the University of Adelaide (Australia) [22], suitable for the design of each study [23,24], were used for the evaluation of the methodological quality and the risk of bias. As a cut-off point for accepting the inclusion of the study in the systematic review, a minimum value of 6 out of 8 was considered for cross-sectional descriptive studies and 6 out of 9 for longitudinal studies. A pilot test was conducted in which each reviewer had to evaluate 3 articles, posteriorly analysing the concordance between their ratings.

## 3. Results

Of the 608 studies initially identified, 26 were selected for systematic review after a critical reading of the full text (Figure 1).

The main characteristics and results obtained in the selected studies are summarized in Table 3.

### 3.1. Description of the Characteristics of the Studies

The number of participants in the studies ranged between 343 and 15,308, over 18 years of age. A total of 72,056 subjects, with the female gender predominating in most of the selected studies. All the articles reviewed analysed the mental health of the adult population as a consequence of restrictive measures to stop the spread of the virus, such as home confinement and physical distancing, with the main emphasis on stress, anxiety, depression and PTSD. Sleep quality and substance abuse were not assessed in this review. Most of the studies (*n* = 24) were cross-sectional, and the other two were longitudinal designs. In terms of geographical distribution, the studies were performed in different regions and countries with very different health systems: China (*n* = 6), Spain (*n* = 3), Germany (*n* = 2), United Kingdom (*n* = 2), Saudi Arabia (*n* = 1), Brazil (*n* = 1), India (*n* = 1), South Korea (*n* = 1), Pakistan (*n* = 1), Jordan (*n* = 1), Italy (*n* = 1), Vietnam (*n* = 1), Turkey (*n* = 1), Bangladesh (*n* = 1) and the US (*n* = 1), noting that two of them were performed in several countries.

To assess the effect of home confinement and social distancing resulting from the SAR-CoV-2 pandemic on the mental health of the general population, different scales and questionnaires were used. The Beck Depression Inventory (BDI), the Short Mood and Feelings Questionnaire (SMFQ), the Patient Health Questionnaire-9 (PHQ-9), the Severity of Dependence Scale (SDS), the Centre for Epidemiologic Studies Depression Scale (CES-D), the PROMIS depression v.8a and the Patient Health Questionnaire-2 (PHQ-2) were used to measure depressive symptoms. The Beck Anxiety Inventory (BAI), the Statistical Anxiety Scale (SAS), the State-Trait Anxiety Inventory (STAI), the PROMIS anxiety v.8a and the Generalised Anxiety Disorder Assessment (GAD-7), the Short version of the Whitely Index and the Health Anxiety Inventory (HAI) for health anxiety were used to assess anxiety. PTSD symptoms were evaluated by the revised Impact of Event Scale-Revised (IES-R), the reduced civilian version of the PTSD checklist (PCL-C-2), the DSM-V PTSD checklist (PCL-5) and the International Trauma Questionnaire (ITQ). The Depression, Anxiety and Stress Scale (DASS) and the Depression, Anxiety and Stress 21-item Scale (DASS-21) were used to evaluate symptoms of anxiety, depression and stress; and the Hospital Anxiety and Depression Scale (HADS) for anxiety and depression. Anxiety, stress and depression levels were measured by the DASS-21 or DASS in nearly half of the studies (*n* = 10), and PTSD was mostly evaluated with IES-R (*n* = 8). GAD-7 and PHQ-9 were also used in numerous studies to evaluate symptoms of anxiety and depression, respectively.

Regarding the statistical analysis, most studies used univariate tests to analyse the effect of sociodemographic and COVID-19-related variables on the main result of the study and multivariate tests to simultaneously analyse various study variables.

When assessing the methodological quality and risk of bias of the studies, most of them obtained high average scores, always above the set cut-off score (Table 4 and Table 5).

### 3.2. Description of the Results

#### 3.2.1. Anxiety Symptoms and Associated Factors

Anxiety symptoms were evaluated in 24 of the 26 studies [25,26,28,29,30,31,32,33,34,35,36,37,38,39,40,41,43,44,45,46,47,48,49,50]. Prevalence differed from 8.3% to 45.1% [25,26,28,32,33,34,36,37,38,39,40,41,42,44,47,49] with the exception of the research conducted by Goularte et al. [32], where 81.1% of the sample reported high levels of anxiety. This variability may be due to the lack of unanimity between the different studies regarding the definition of anxiety or the established cut-off point. In Massad et al.’s study [38], mild anxiety was reported in 21.5%, moderate anxiety in 10.9% and severe anxiety in 6% of participants; Özdin et al. [41] found symptoms of anxiety, in general, in 45.1% of the sample. Benke et al. [28] demonstrated a prevalence of 29.4% for anxiety and 21.1% of the sample obtained above cut-off point levels of anxiety disorder.

Many factors were associated with higher levels of anxiety during the COVID-19 pandemic. Women were more likely to develop anxiety symptoms compared to men, with the exception of data provided by Chen et al. [29] and Wang et al. [49], which indicated a higher incidence in males. Younger ages were also associated with anxiety [25,28,29,30,34,35,36,37,43,44,45,46,47,49]. The student [26,33,35], unemployed [28,35,44], housewife [33] or health worker status [33,35] reported more anxiety compared to other occupational status (worker, retired people, etc.). Some studies also associated lower income [29,31,32,35,37,46,47,48], education [28,29,32,33,47] and the perception of the information received about the pandemic [31,37] with anxious symptomatology. Participants with a history of mental illness or current or prior psychiatric treatment reported being more anxious than healthy people [28,32,33,41,45,46]. A study also associated living alone with anxiety compared to subjects living with dependents (spouse, children, family members, caregivers) [28]. Widespread linear models linked lower levels of anxiety to factors such as feeling healthy, high incomes and a broad social network and social support [29,38]. Conversely, sociodemographic variables such as being female, young, student, divorced or widowed, having low levels of education and income, feelings of loneliness, suffering from previous psychiatric illness or having a history of mental illness and worse self-perceived health were considered the main factors that are associated with anxious symptomatology [30,31,32,37,38,39,41,44,45,46,47,48]. Regarding the variables related to the COVID-19, high concern about the pandemic, social distancing measures and perception of risk were related to anxiety [26,30,31,37,44,46].

González-Sanguino et al. [31] identified misinformation as one of the main factors that are associated with anxiety, while Lei et al. [37] demonstrated that people with more knowledge related to the COVID-19 were more likely to experience anxiety during the pandemic. The frequency of news consumption about the COVID-19 [42] and the dissemination of health information about the pandemic over radio [50] were also associated with higher scores.

Finally, taking into consideration the area of residence of the people, Özdin et al. [41] stated that living in urban areas contributed to greater anxiety, while Schweda et al. [45] found living in rural areas was associated with anxiety.

#### 3.2.2. Depressive Symptoms and Associated Factors

Symptoms of depression were evaluated in 23 of the 26 studies, with a prevalence from 14.6% to 46.42% [25,26,27,28,30,32,33,34,35,36,37,39,40,41,43,44,46,47,48,49,50]. Research by Goularte et al. [32] and Ripon et al. [43] demonstrated some signs of depression in 81.9% and 85.9% of the population studied, respectively. Most studies [25,26,28,30,31,32,34,35,37,42,43,45,46,47,49] associated young age and female sex or gender with greater depressive symptoms than men, and women also reported greater symptomatology compared to men [26,28,31,32,35,36,37,42,43,45,47], except in the study of Wang et al. [49] where men were the ones who showed the worst results. A low-income level contributed significantly to worse mental health, with low- or non-income people, such as students or unemployed individuals, exhibiting the most depressive symptoms [28,31,32,33,40,43,46,47,48,49]. As for marital status, singles, divorcees or widowers, and people living alone found themselves more depressed than married people and couples [32,33,36,37,43,46].

As in the case of anxiety, people with low levels of education were more likely to develop depressive symptoms than people with higher education levels [28,31,32,33,46,47,49], except for Ripon et al. [43], who reported worse depression outcomes in people with higher educational level. Being or having been in psychiatric treatment and presenting mental health problems was also associated with the onset of depressive symptoms during home confinement and social isolation derived from the pandemic [28,31,32,33,39,40,46]. Multivariate analyses showed that the main factors related to sociodemographic variables were being a female, young, having lower levels of education and income, or student, unemployed or housewife status; being a widower, divorcer or unmarried person; having feelings of loneliness; having previous psychiatric illness and worse self-perceived health [30,31,32,37,39,40,46,47,48,49]. Pandemic-related variables such as concern for the COVID-19, lack of psychological support, risk perception and long periods of social distancing also contributed to the onset of depressive symptoms [30,31,32,37,39]. Protective factors, such as spiritual well-being [31], being over 60 and having a partner [39], were identified.

#### 3.2.3. Stress Symptoms, PTSD and Associated Factors

Of the total studies, 10 analysed stress levels [26,30,33,36,39,40,42,44,49,50] and 9 PTSD-related symptoms [26,31,32,42,43,46,47,49,50]. The prevalence of stress-related symptoms and PTSD differed from 8.1% to 49.66% [26,31,32,33,36,39,40,43,46,48,49]; but Ripon et al. [43] claimed to find symptoms of PTSD in 81.8% of the participants, of whom only 20% reported a likely diagnosis of PTSD. In his longitudinal study, Wang et al. [50] observed a significant increase in PTSD levels over time. Most research associated the female gender with higher levels of stress [26,31,32,36,39,43,49], but three of the studies found greater symptomatology in men compared to women [42,46,49]. Younger people generally showed more stress [26,30,31,32,39,42,43,44,47], but Lee et al. [36] found worse results among older people. People without income, such as students, housewives or those unemployed, proved to be more susceptible to develop symptoms of PTSD and stress than those with a job and income [26,31,32,33,40,48]. In terms of educational level, most studies reported that lower levels correlated with higher stress and PTSD [31,32,33,49], but Ripon et al. [43] found greater symptomatology among people with higher educational levels. All in all, the main factors that contributed to PTSD and stress were being a female, young, having feelings of loneliness, a low level of education and income, a student or unemployed status and previous psychiatric illness [30,31,32,39,40,43,46,47,49]. Regarding pandemic-related factors, concern about the COVID-19, social distancing, perception of danger and receiving insufficient information were the main factors associated with stress and PTSD [30,31,32,47,49].

## 4. Discussion

Considering a global perspective, the main objective of this systematic review was to explore the mental health of the general population, in terms of depression, stress and anxiety, during social distancing and home confinement resulting from the SARS-CoV-2 outbreak. This review revealed the main factors that are associated with the development of anxiety, depression and PTSD during the pandemic, being females, young age, unemployed and people with previous mental health or psychiatric illnesses the most vulnerable.

Women developed higher levels of anxiety, stress and depression, but men also demonstrated some risk of experiencing elevated stress levels and PTSD during the COVID-19 [29,46,49]. Although women are at a lower risk of experiencing severe symptoms or even dying due to SARS-CoV-2 than men [51], they have been shown to be more vulnerable to the psychological consequences of the pandemic [52].

Younger people proved to be more vulnerable to the development of health-related symptoms [26,30,31,32,39,42,43,44,47], anxiety [25,28,29,30,34,35,36,37,43,44,45,46,47,49] and depression [25,26,28,30,31,32,34,35,37,42,43,45,46,47,49]. The study conducted by Glowacz and Schmits [53] to assess COVID-19-related psychological discomfort by age showed that young people between 18 and 30 were the most psychologically affected by the pandemic.

Loss of income during the COVID-19 crisis has been shown to have harmful implications for mental health [54] and to be an important factor associated with poor mental health in times of social-health crisis [55]. Absence or decrease in income promotes the appearance of anxiety [29,31,32,35,37,46,47,48], stress [26,31,32,33,40,48] and depression [28,31,32,33,40,43,46,47,48,49], with the unemployed, housewives and students being the most affected groups during the pandemic.

Additionally, having a previous psychiatric illness or being treated for a psychological disorder was also associated with mental health disturbances during periods of social distancing and home confinement [28,30,31,32,33,37,39,40,41,43,46,47,48,49].

The perception of insufficient information was a very common factor associated with poor mental health in the revised studies, but receiving negative data has also shown high levels of anxiety [31,42,50] and stress [30,31,32,47,49], as well as concern for the pandemic. Although previous pandemic-related research showed a significant association between receiving enough information and good mental health, it has never been possible to communicate as quickly or access such immense amounts of information in real-time as today [56]. Responsible use of information dissemination and acquisition tools can help to spread scientific findings, share protocols of action and diagnosis, compare different approaches globally, provide psychological support to the population, etc. [57]; however, access to these huge amounts of information do not always involve the acquisition of reliable data, and some people may not be in a position or have the necessary skills to properly process and understand the information received [58]. Sharing fake news, conspiracy theories, magic cures and other decontextualized news increases the anxiety and stress of the population [59]. The WHO created the term “infodemic” to refer to the phenomenon by which an excessive amount of information about a problem is accessed and may be associated with inadequate public health responses and create confusion and mistrust among the population [60]. Proper management of information through reliable and verifiable sources, avoiding decontextualized use, is essential for the population to understand and adapt to the health measures dictated by the authorities [61].

The long-term implications of the previously mentioned mental health disturbances are a cause for concern. While the female gender, younger ages, lack or decrease in income, previous psychiatric illness and perception of lack of information or “infodemic” are the main factors associated with symptoms of depression, anxiety and stress, social networks and economic stability have been shown to be the main factors associated with good mental health during the pandemic [29,38,40].

All of these aspects should be considered when making government changes and implementing measures to reduce symptoms of anxiety, stress and depression. Social support and a proactive approach to the most vulnerable groups such as women, young people and people with previous mental illnesses could lead to the prevention, early detection and intervention of mental health symptoms arising from the COVID-19 pandemic [52]. Being clear about who are the people most affected by physical and social distancing measures during the pandemic facilitates the design of specific personalized self-care strategies and allows providing guidance and assistance to the most vulnerable groups of people to help and support their well-being.

To lessen the fear of a new recession and financial collapse, apart from strong and resilient leadership of the authorities, medium- and long-term planning is needed to rebalance and revive the economy [62]. In its plan for the global management of the SARS-CoV-2 pandemic, the United Nations Educational, Scientific, and Cultural Organization (UNESCO) highlighted the positive effect of developing clear, direct and timely information policies on the behaviour of citizens, dedicating two of its eleven ethical considerations to this subject [63]. Social media and television are the most widely used sources of information, but there is a need to improve the quality and veracity of Internet information for the general population [64]. Currently, most of the population in developed countries has access to the internet or the media, which should be used to provide specific recommendations and direct citizens to official and reliable sources.

This review provides information that will guide future research and facilitate the development of programs to alleviate pandemic-derived mental health symptoms. Knowledge of the impact of COVID-19 grows daily, so studies related to the pandemic must be continuously updated. For future investigations, these findings should be interpreted considering the limitations of this study and the revised research. The fact that all studies related to this topic were performed during the SARS-CoV-2 outbreak, the randomization of the sample may not be possible in some cases. Additionally, the online survey was the main method of information collection, being able to trigger biases in the selection of participants, such as oversampling of people with higher levels of education or younger. The absence of consideration of response rates and recruitment processes in the reviewed studies could also bias the results obtained. By removing review studies that focus on specific populations, such as health workers, people with specific diseases or the elderly, key findings related to particularly vulnerable and underrepresented communities may have been missed. Most of the studies included in the review were cross-cutting and may have assessed mental health at different stages of the outbreak, making it difficult to establish causal associations between the pandemic and the levels of depression, stress and anxiety of the population. Furthermore, although all governments based their policies on physical distancing, the measures were not the same in all countries, which may also influence the findings of the studies. Another considerable limitation is heterogeneity in the criteria used in the different studies reviewed to consider the presence of anxiety, stress and depression.

Regarding its strengths, this study identifies the most vulnerable groups of the general population, highlighting the differences between groups and identifying the factors that are associated with worse mental health. In addition, factors such as the presence of a broad social support network, receiving enough and quality information and having economic resources become particularly relevant at difficult times such as the COVID-19 social-health crisis. This scientific evidence facilitates the development of programs based on the specific needs of the general population to reduce the impact of the pandemic on its levels of stress, anxiety and depression.

## 5. Conclusions

The COVID-19 pandemic raised public health concerns not only in terms of physical health but also associated with a number of mental health disturbances. This review demonstrated that females, young age, unemployed, and patients with previous mental health or psychiatric illnesses were the most vulnerable. When disseminating information on the pandemic, governmental organisations, authorities and other professionals must use reliable and dependable sources to avoid misinformation or information overload and reduce mental health behaviours.

## Figures and Tables

**Figure 1 ijerph-18-06528-f001:**
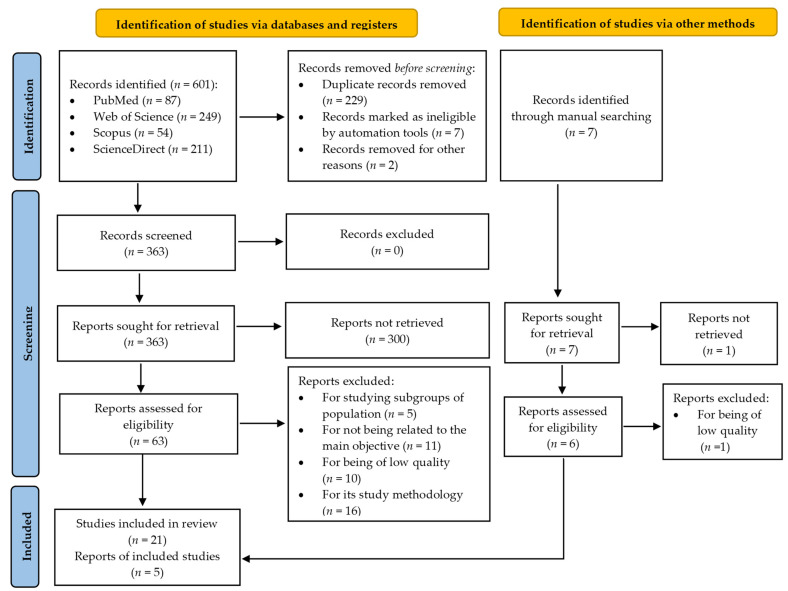
Study selection flowchart.

**Table 1 ijerph-18-06528-t001:** PIO format: keywords.

**Population**	General adult population
**Intervention**	Measure the effect of social distancing and home confinement resulting from the COVID-19 pandemic on mental health
**Outcomes**	Depression level, stress level and anxiety level
**Research Question**	Do the social distancing and home confinement regulations resulting from the SAR-CoV-2 pandemic have repercussions on the mental health of the general population, affecting their levels of anxiety, stress or depression?

**Table 2 ijerph-18-06528-t002:** Search strategy used, adapted to each of the databases.

Database	Search Strategy
Pubmed	(“sars virus”(MeSH Terms) OR “sars virus”(Title/Abstract) OR “SARS-Cov-2”(MeSH Terms) OR “SARS-Cov-2”(Title/Abstract) OR “pandemic”(Title/Abstract) OR “severe acute respiratory syndrome coronavirus”(Title/Abstract) OR “COVID-19”(Title/Abstract)) AND (“mental health”(MeSH Terms) OR “mental health”(Title/Abstract) OR “Psychological health”(Title/Abstract)) AND (“home confinement”(Title/Abstract) OR “physical distancing”(MeSH Terms) OR “physical distancing”(Title/Abstract)) AND (“adult”(MeSH Terms) OR “adult”(Title/Abstract) OR “general population”(Title/Abstract) OR “general public”(Title/Abstract) OR “public”(Title/Abstract) OR “community”(Title/Abstract))
Web of Science	TS = (sars virus OR SARS-Cov-2 OR pandemic OR severe Acute Respiratory Syndrome Coronavirus OR COVID-19) AND TS = (mental health OR Psychological health) AND TS = (home confinement OR Physical Distancing) AND TS = (adult OR general population OR general public OR public OR community)
Scopus	TITLE-ABS-KEY ((“sars virus”) OR (SARS-Cov-2) OR (pandemic) OR (“severe Acute Respiratory Syndrome Coronavirus”) OR (COVID-19)) AND TITLE-ABS-KEY ((“mental health”) “ OR (“Psychological health”)) AND TITLE-ABS-KEY ((“home confinement” OR “Physical Distancing”)) AND TITLE-ABS-KEY ((adult) OR (“general population”) OR (“general public”) OR (public) OR (community))
Science Direct	(“sars virus” OR SARS-Cov-2 OR pandemic OR “severe Acute Respiratory Syndrome Coronavirus” OR COVID-19) AND (“mental health” OR “Psychological health”) AND (“home confinement” OR “Physical Distancing”) AND (adult OR “general population” OR “general public” OR public OR community)

**Table 3 ijerph-18-06528-t003:** Characteristics of the studies included in the systematic review.

Study/Author	Typology/Main Objective	Participants	Variables/ Instruments	Main Findings	JBI
Ahmed et al. [25], 2020	Design: Descriptive cross-sectional, Objective: To study the psychological morbidity induced by the COVID-19 pandemic.	*n* = 1074 Age: >18 years Sex (f/m): 503/571	Anxiety: BAI Depression: BDI	In total, 29% suffered high levels of anxiety, and 37.1% presented different forms of depression. The proportion of people with different levels of anxiety (*p* < 0.001) and depression (*p* < 0.001) was significantly higher in the age group 21–30 years.	6/8
Alkhamees et al. [26], 2020	Design: Quantitative cross-sectional Objective: To assess the psychological impact of the COVID-19 pandemic during the curfew and closure.	*n* = 1160 Age: >18 years Sex (f/m): 741/419	Anxiety: DASS-21 Depression: DASS-21 Stress: DASS-21 PTSD: IES-R	In total, 28.3%, 24% and 22.3% reported moderate or severe depression, anxiety and stress, respectively. The female sex, the age 18–40, and being a student were significantly associated with higher levels of PTSD, anxiety, depression and stress (*p* < 0.05). Experiencing shortness of breath and dizziness showed a strong association with levels of anxiety, stress and depression (*p* < 0.001). Social distancing decreased stress and anxiety (*p* < 0.05), while hand hygiene decreased depression (*p* < 0.05).	8/8
Ammar et al. [27], 2020	Design: Quantitative cross-sectional Objective: To analyse the impact of COVID-19 restrictions on mental health and emotional well-being.	*n* = 1047 Age: >18 years Sex (f/m): 563/484	Depression: SMFQ	A significant change was observed in mood, well-being and feelings (*p* < 0.001); participants showed more depressive symptoms during home confinement in relation to previous moments.	7/8
Benke et al. [28], 2020	Design: Quantitative cross-sectional Objective: To identify predictors of worse mental health during the COVID-19 pandemic.	*n* = 4335 Age: >18 years Sex (f/m): 3284/1051	Generalized Anxiety: GAD-7 Health Anxiety: Short Version of the Whitely Index Depression: PHQ-9	In total, 31.1% exceeded the cut-off score for a depression diagnosis, 21.2% for anxiety disorder and 29.4% for health anxiety. Women reported more anxiety and depression than men. Being young, low educational level, unemployment, current or previous psychiatric treatment, belonging to a risk group, anguish related to the restriction of social contacts, and a greater perception of change predicted depression and anxiety (*p* < 0.001). Living alone also contributed to increased anxiety.	8/8
Chen et al. [29], 2021	Design: Quantitative cross-sectional Objective: To compare the anxiety levels of confined people with those who were not confined during the second wave of the COVID-19 pandemic.	*n* = 1837 Age: >18 years Sex (f/m): 1512/325	Anxiety: STAI	Severe anxiety increased in participants aged between 26 and 39 years, in men, in people with low incomes and in those with a level of education below a bachelor´s degree (*p* < 0.001). Participants who had a general feeling of good health and were not in quarantine showed less anxiety than those who felt they were in poor health and were in quarantine (*p* < 0.001). Furthermore, high income was an independent protective factor for anxiety (*p* = 0.027), and a poor state of health was an independent risk factor (*p* < 0.001).	8/8
Dean et al. [30], 2021	Design: Quantitative cross-sectional Objective: To examine the psychosocial distress during the initial phase of the pandemic in four different societies.	*n* = 1306 Age: >18 years Sex (f/m): 904/400	Anxiety: DASS Depression: DASS Stress: DASS	Younger age (β = −0.13; t = −2.98; *p* = 0.002), greater concern about COVID−19 (β = 0.15; t = 3.01; *p* = 0.003) and greater feelings of loneliness (β = −0.23; t = 8.20; *p* < 0.001) predicted a worse psychological outcome, but the magnitude of these effects varied among the four regions.	8/8
González-Sanguino et al. [31], 2020	Design: Quantitative cross-sectional Objective: To analyse the psychological impact of the COVID-19 outbreak three weeks after the outbreak of the pandemic and declaration of the state of alarm.	*n* = 3480 Age: >18 years Sex (f/m): 2610/870	Generalized anxiety: GAD-7 Depression: PHQ-9 PTSD: PCL-C2	In total, 18.7% presented depressive symptoms, 21.6% anxiety and 15.8% PTSD. Female sex, previous mental health problems, symptoms associated with the virus or those with an infected close relative were associated with the worst results in the three variables (*p* < 0.05), and age, economic stability and the information received about the pandemic were negatively correlated with symptoms (*p* < 0.05). Spiritual well-being was a protective factor for depression and being a student or feelings of loneliness were risk factors (*p* < 0.001). Low spiritual well-being, feelings of loneliness, being a woman and not enough information predicted higher anxiety (*p* < 0.001) and post-traumatic stress (*p* < 0.001).	8/8
Ferraz-Goularte et al. [32], 2020	Design: Quantitative cross-sectional Objective: To investigate the prevalence and determinants of psychiatric symptoms during the COVD-19 pandemic.	*n* = 1996 Age: >18 years Sex (f/m): 1676/320	Anxiety: PROMIS anxiety v.8ª Depression: PROMIS depression v.8ª PTSD: IES-R	Anxiety (81.9%) and depression (68%) were the most frequent psychiatric symptoms, and 34.2% of the participants reported PTSD. Female sex, longer duration of social distancing measures and previous psychiatric illness were significantly associated with higher levels of stress, depression and anxiety (*p* < 0.01). Furthermore, young age, low education and/or income were correlated with greater symptoms (*p* < 0.01). Being single was only associated with greater depression and anxiety (*p* < 0.01).	8/8
Hazarika et al. [33], 2021	Design: Quantitative cross-sectional Objective: To evaluate the psychological state during the initial phase of the confinement produced by COVID-19.	*n* = 422 Age: >18 years Sex (f/m): 255/167	Anxiety: DASS-21 Depression: DASS-21 Stress: DASS-21	In total, 35.5% reported stress, 32% anxiety and 34.7% depression. Single people, students, housewives, people who work in the public sector, people with a history of mental illness and those with lower educational levels were shown to be more likely to experience symptoms of stress, anxiety and depression (*p* < 0.05).	6/8
Huang et al. [34], 2020	Design: Quantitative cross-sectional Objective: To assess the mental health burden during the COVID-19 outbreak and explore possible influencing factors.	*n* = 7236 Age: >18 years Sex (f/m): 3952/3284	Generalized anxiety: GAD-7 Depression: CES-D	The overall prevalence of generalized anxiety disorder and depression was 35.1% and 20.1%, respectively. Younger people (<35 years) and those who spent 3 h or more/day thinking about COVID-19 reported a significantly higher prevalence of generalized anxiety disorder and depression (*p* < 0.05).	7/8
Lal et al. [35], 2020	Design: Quantitative cross-sectional Objective: To evaluate the psychological distress caused by the pandemic of the COVID-19 disease.	*n* = 1000 Age: >18 years Sex (f/m): 427/573	Generalized anxiety: GAD-7 Depression: PHQ-9	Women reported more depression (*p* < 0.001) and anxiety (*p* = 0.03) than men. In addition, participants between 18 and 30 years of age disclosed greater anxiety (*p* = 0.001) and depression (*p* = 0.004). Lower-income individuals, students, healthcare workers and unemployed also showed worse results (*p* < 0.05).	6/8
Lee et al. [36], 2021	Design: Quantitative cross-sectional Objective: To assess mental health and social well-being during the COVID-19 pandemic.	*n* = 400 Age: >18 years Sex (f/m): 110/287	Anxiety: DASS Depression: DASS Stress: DASS	Depression was present in 36.75% of the participants, anxiety in 29.5% and stress in 24.5%. The youngest reported feeling more worried, anxious or tense (*p* = 0.04), while the oldest reported higher levels of stress (*p* = 0.02). Women reported poorer mental health in general (*p* = 0.001), except for anxiety, where there were no significant differences between groups. Singles showed greater depression than those married or with a partner (*p* = 0.03).	6/8
Lei et al. [37], 2020	Design: Quantitative cross-sectional Objective: To evaluate and compare the prevalence and associated factors of anxiety and depression during the COVID-19 outbreak.	*n* = 1593 Age: >18 years Sex (f/m): 976/617	Anxiety: SAS Depression: SDS	The prevalence of anxiety and depression was 8.3% and 14.6%, respectively. Female gender and age ≤ 30 years old were associated with greater symptoms of depression and anxiety (*p* < 0.05). Having knowledge about COVID-19 (β = 0.621; *p* = 0.032), economic losses (β = 0.634; *p* = 0.001), being divorced or widowed (β = 4.825; *p* = 0.001), bad self-perceived health (β = −2.762; *p* < 0.001) and concern about infection (β = 1.62; *p* < 0.001) predicted more anxiety. Absence of psychological support (β = 1.327; *p* = 0.043), being divorced or widowed (β = 7.313; *p* < 0.001), bad self-perceived health (β = −3.109; *p* < 0.001) and greater concern about infection (β = 1.232; *p* = 0.006) were associated with depression.	8/8
Massad et al. [38], 2020	Design: Quantitative cross-sectional Objective: To evaluate the prevalence of psychological distress related to quarantine and to explore sociodemographic correlations.	*n* = 5274 Age: >18 years Sex (f/m): 2914/2360	Anxiety: BAI	The prevalence of mild, moderate and severe anxiety was 21.5%, 10.9% and 6%, respectively. Female gender or the presence of more members in the household were correlated with higher levels of anxiety; old age, a large social network, social support and high income correlated with lower levels.	8/8
Mazza et al. [39], 2020	Design: Cross-sectional quantitative Objective: To establish the prevalence of psychiatric symptoms and to identify risk and protective factors for psychological distress.	*n* = 2766 Age: >18 years Sex (f/m): 1982/784	Anxiety: DASS-21 Depression: DASS-21 Stress: DASS-21	In total, 17% reported a high level of depression and 15.4% an extremely high range. Regarding anxiety, 7.2% had a high level, and 11.5% were in the extremely high range. Regarding stress, 14.6% were in the high range, and 12.6% were in an extremely high range. Female sex, having family members with COVID-19, negative affect and detachment were associated with higher levels of depression, anxiety and stress (*p* < 0.05). Additionally, previous medical problems were associated with higher levels of depression and anxiety, and younger people reported more stress (*p* < 0.05).	8/8
Ngoc Cong Duong et al. [40], 2020	Design: Quantitative cross-sectional Objective: To estimate the prevalence of psychological problems and identify the factors associated with the psychological impact of COVID-19 during the first national blockade.	*n* = 1412 Age: >18 years Sex (f/m): 532/880	Anxiety: DASS-21 Depression: DASS-21 Stress: DASS-21 PTSD: IES-R	In total, 23.5% experienced depression, 14.1% anxiety and 22.3% stress. People aged ≥60 years demonstrated lower levels of depression, and unemployed people, students, housewives and people with chronic diseases had a higher risk of depression. Isolated participants were more likely to experience anxiety, and unemployed people or students reported higher levels of stress.	8/8
Özdin et al. [41], 2020	Design: Quantitative cross-sectional Objective: To assess the levels of depression, anxiety and anxiety about health during the COVID-19 pandemic and examine the factors that affect them.	*n* = 343 Age: >18 years Sex (f/m): 169/174	Anxiety: HADS Health anxiety: HAI Depression: HADS	In total, 23.6% scored above the cut-off point for depression and 45.1% for anxiety. Living in urban areas was associated with higher levels of depression and anxiety. Female gender (β = 0.105; *p* = 0.047), suffering from a chronic illness (β = 0.160; *p* = 0.003) and having a previous psychiatric illness (β = 0.176; *p* = 0.001) were risk factors for predicting health anxiety.	8/8
Panchuelo-Gómez et al. [42], 2020	Design: Quantitative longitudinal Objective: To evaluate the temporal evolution of the psychological impact of the crisis and closure of COVID-19.	*n* = 4724 Age: >18 years	Anxiety: DASS-21 Depression: DASS-21 Stress: DASS-21 PTSD: IES-R	Anxiety, depression and stress levels were significantly higher over time, with a prevalence of 37.22%, 46.42% and 49.66%, respectively. More anxiety and stress were found in younger people, and more depression in single subjects. The frequency of consumption of news about COVID-19 was a factor clearly associated with higher levels of anxiety, depression and stress.	7/9
Ripon et al. [43], 2020	Design: Quantitative cross-sectional Objective: To assess the prevalence of depression and PTSD among quarantined people during the COVID-19 outbreak.	*n* = 5792 Age: >18 years	Depression: CES-D PTSD: IES-R	In total, 85.9% reported depressive symptoms and 81.8% PTSD, of which 20% had a probable diagnosis of PTSD, and 24.3% demonstrated PTSD as a clinical problem. Depression and PTSD were more frequent in people aged 31–45 years, with low income, with higher education, single and in-home quarantine (*p* < 0.05). Women showed higher levels of depression, while PTSD was more frequent in men (*p* < 0.05).	6/8
Rodríguez-Rey et al. [44], 2020	Design: Cross-sectional quantitative Objective: To explore the mental health during the early stages of the COVID-19 outbreak.	*n* = 3055 Age: >18 years Sex (f/m/o): 2293/744/18	Anxiety: DASS-21 Depression: DASS-21 Stress: DASS-21 PTSD: IES-R	In total, 36% of the participants reported moderate to severe psychological impact, 25% mild to severe anxiety levels, 41% depressive symptoms, and 41% felt stressed. Women, young people and those who lost their jobs during the pandemic had worse results (*p* < 0.05). A higher self-perceived health was associated with less anxiety and depression (*p* < 0.001), and doing leisure activities during the day reduced stress, anxiety and depression (*p* < 0.001).	7/8
Schweda et al. [45], 2020	Design: Quantitative cross-sectional Objective: To investigate the psychological reactions in response to real or perceived threats of SARS-Cov-2 infection.	*n* = 15308 Age: >18 years Sex (f/m/o): 10824/4433/51	Generalized anxiety: GAD-7 Depression: PHQ-2	Women, young people, those residing in rural areas, people with previous psychiatric illness and who did not trust government actions against COVID-19 reported higher levels of anxiety (*p* < 0.001).	8/8
Sherman et al. [46], 2020	Design: Quantitative cross-sectional Objective: To examine some of the burdens of the pandemic, the prevalence of mental health problems and the risk factors for psychosocial morbidity.	*n* = 591 Age: >18 years Sex (f/m): 458/133	Generalized anxiety: GAD-7 Depression: PHQ-9 PTSD: PCL-5	Young people, women and participants with lower incomes were more likely to have depression and anxiety (*p* < 0.005), and people with a lower educational level suffered from depressive symptoms (*p* = 0.005). Depression was associated with previous mental illness (*p* < 0.0001), not being married (*p* = 0.008) and a greater alteration in daily life (*p* < 0.001). Higher levels of anxiety were linked to younger age (*p* < 0.005), previous mental illness (*p* < 0.0001) and greater disruption in daily life (*p* < 0.001). PTSD was associated with previous mental illness (*p* < 0.001) and greater disruption in daily life (*p* < 0.0002).	8/8
Shevlin et al. [47], 2020	Design: Quantitative cross-sectional Objective: To investigate the prevalence of symptoms of anxiety, generalized anxiety, depression and trauma related to COVID-19 during an early phase of the pandemic and to estimate associations with variables.	*n* = 2025 Age: >18 years Sex (f/m/o): 1047/972/6	Generalized anxiety: GAD-7 Depression: PHQ-9 PTSD: ITQ	In total, 22.1% had depression symptoms, 21.6% anxiety and 16.79% PTSD. In the case of PTSD, there was a significant gender difference, with a higher prevalence in men (*p* < 0.01), the same situation that was observed with anxiety, but in this case with women (*p* < 0.01). Symptoms of anxiety, depression and PTSD were predicted by a young age, children at home and elevated risk perception. Low or loss of income and previous health problems also predicted anxiety and depression.	8/8
Smith et al. [48], 2020	Design: Cross-sectional quantitative Objective: To assess the impact of social distancing during COVID-19 on mental health.	*n* = 932 Age: >18 years Sex (f/m/o): 590/334/8	Anxiety: BAI Depression: BDI	The prevalence of poor mental health due to the pandemic was 36.8%. Female sex, aged 25–34 years, a lower annual income, smoke and suffering from physical multimorbidity were associated with higher levels of anxiety and depression (*p* < 0.05).	8/8
Wang et al. [49], 2020	Design: Cross-sectional quantitative Objective: To establish the prevalence of psychiatric symptoms and to identify risk and protector factors of psychological stress.	*n* = 1210 Age: >18 years Sex (f/m): 814/396	Anxiety: DASS-21 Depression: DASS-21 Stress: DASS-21 PTSD: IES-R	In total, 16.5% showed moderate to severe depressive symptoms, 28.8% moderate to severe anxiety symptoms and 8.1% moderate to severe stress. Men had less PTSD but greater symptoms of anxiety, depression and stress (*p* < 0.05), and students demonstrated more PTSD, stress and anxiety (*p* < 0.05). Contact with an infected person or material was shown to be a risk factor for anxiety and depression (*p* < 0.01). People with lower educational levels had greater depressive symptoms (*p* < 0.01), and dissatisfaction with the amount of information received was associated with greater stress (*p* < 0.05).	8/8
Wang et al. [50], 2020	Design: Quantitative longitudinal Objective: To assess the temporary psychological impact and adverse mental health status during the initial and peak outbreak of the COVID-19 pandemic and identify risk and protective factors.	*n* = 1738 Age: >18 years	Anxiety: DASS-21 Depression: DASS-21 Stress: DASS-21 PTSD: IES-R	PTSD increased over the time (*p* <0.01), but not the levels of anxiety, depression and stress (*p* > 0.05). Younger participants demonstrated higher levels of PTSD (B = 0.77, t = 2.28, *p* <0.05) and subjects who lived in a household with 3–5 people (B = 1.32, t = 2.04, *p* < 0.05) and more than 6 people (B = 1.44, t = 2.20, *p* <0.05) reported more PTSD compared to people who lived alone. Radio broadcast of information about COVID-19 was associated with higher anxiety and depression scores (*p* < 0.05).	7/9

JBI: Total score in the Joanna Briggs Institute “Critical Appraisal Tools”; *n*: number of participants; sex (f/m): sex (female/male).

**Table 4 ijerph-18-06528-t004:** The results of the quality assessment of quasi-experimental studies.

Study	JBI	Q1	Q2	Q3	Q4	Q5	Q6	Q7	Q8	Q9
Panchuelo-Gómez et al. [42]	7/9	+	+	−	−	+	+	+	+	+
Wang et al. [50]	7/9	+	+	−	−	+	+	+	+	+

JBI: Joanna Briggs Institute; Q: question.

**Table 5 ijerph-18-06528-t005:** The results of the quality assessment of cross-sectional quantitative studies.

Study	JBI	Q1	Q2	Q3	Q4	Q5	Q6	Q7	Q8
Ahmed et al. [25]	6/8	+	+	+	+	−	−	+	+
Alkhamees et al. [26]	8/8	+	+	+	+	+	+	+	+
Ammar et al. [27]	7/8	+	+	+	+	+	−	+	+
Benke et al. [28]	8/8	+	+	+	+	+	+	+	+
Chen et al. [29]	8/8	+	+	+	+	+	+	+	+
Dean et al. [30]	8/8	+	+	+	+	+	+	+	+
González-Sanguino et al. [31]	8/8	+	+	+	+	+	+	+	+
Goularte et al. [32]	8/8	+	+	+	+	+	+	+	+
Hazarika et al. [33]	6/8	+	+	+	+	−	−	+	+
Huang et al. [34]	7/8	+	+	+	+	+	−	+	+
Lal et al. [35]	6/8	+	+	+	+	−	−	+	+
Lee et al. [36]	6/8	+	+	+	+	−	−	+	+
Lei et al. [37]	8/8	+	+	+	+	+	+	+	+
Massad et al. [38]	8/8	+	+	+	+	+	+	+	+
Mazza et al. [39]	8/8	+	+	+	+	+	+	+	+
Ngoc Cong Duong et al. [40]	8/8	+	+	+	+	+	+	+	+
Özdin et al. [41]	8/8	+	+	+	+	+	+	+	+
Ripon et al. [43]	6/8	+	+	+	+	−	−	+	+
Rodríguez-Rey et al. [44]	7/8	+	+	+	+	+	−	+	+
Schweda et al. [45]	8/8	+	+	+	+	+	+	+	+
Sherman et al. [46]	8/8	+	+	+	+	+	+	+	+
Shevlin et al. [47]	8/8	+	+	+	+	+	+	+	+
Smith et al. [48]	8/8	+	+	+	+	+	+	+	+
Wang et al. [49]	8/8	+	+	+	+	+	+	+	+

JBI: Joanna Briggs Institute; Q: question. + favorable score on the question; − unfavorable score on the question

## Data Availability

Not applicable.

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
