# Peer review of "Psychological Effects of Home Confinement and Social Distancing Derived from COVID-19 in the General Population—A Systematic Review"

_ijerph, 2021, doi:10.3390/ijerph18126528_

Round 1
Reviewer 1 Report
It is meaningful enough as a review paper. However, it is necessary to consider the following points.
First, it seems to analyze by adopting a general meta analysis method.
Second, it is currently described focusing on only three things: anxiety, depression, and stress. Other negative psychological conditions need to be described.
Third, it is necessary to describe the difference between the results from this studies, mainly focusing the context of COVID-19 and other previous studies on anxiety, depression, and stress.
Author Response
Consulte el archivo adjunto.

Reviewer 2 Report
This is a really important study providing a review of the published research (up to Jan 2021), carried out during the Covid-19 pandemic to understand the impact on mental health. The introduction, methods and results sections are written very well they are extremely clear and well described. However the discussion is notably different in writing style and requires proof reading for English language errors (both grammar and incorrect words being used), as well as re-writing for other issues described below.
I think this paper should be accepted for publication once the following have been addressed:
- The main aim of the study and the search terms are not entirely clear to me. If the review is limited to studies that looked at the impact of home confinement and mental health, which the objectives and search terms suggest, this needs really clearly defining as a concept. Does it mean anyone who might have had restrictions on their lives or might not have had many at all (i.e. those who still worked?); or was the research on people who definitely were confined when the research was completed? In addition, the results talk about a wide range of factors that were associated with depression/anxiety, but were all of these in addition to social isolation / home confinement or as well as?
- Greater clarity on the above, and clearer explanation in the results section would be very helpful.
Methods:
- The focus population was "general population" which is reasonable, but by dropping studies focussed on specific populations there will be key findings missed that relate to more vulnerable, under-represented communities. This needs to be highlighted as a limitation in the discussion.
- Countries had different levels of measures and at different time points so this also needs clarifying in the methods section - was this taken into consideration in the acceptance of the papers into the reviews?
- The review does not mention consideration of the response rates and recruitment processes for these studies - these could have created bias in the findings.
- The results section notes how varied definitions of mental health were. Is it possible at all to seperate results by levels of severity of mental health? For example it would be good to look at 'Clinically relevant' symptoms which would include moderate / severe but not mild. If you can't do this, it is a weakness of the study that needs highlighting in the discussion.
Discussion
- The discussion includes a lot of hypotheses / supposition about the causes of the identified factors associated with mental ill health. However none of these are based on findings from the review and i am concerned that they are potentially misleading. I would recommend that the authors focus on summarising the findings from the review only without the need to explain why each of those things might be related to mental health.
- The discussion claims that (page 5 line 335) government responses largely caused mental ill health. This wasn't mentioned in the results section and relates to a single referecne, i don't think this is something that the findings of this review can claim. Other similar claims should be reviewed and removed from the discussion.
- The section on implications for policy and practice on page 6 is useful and could be the main focus for the discussion.
- The strengths and limitations section needs expanding upon. I have noted a few other limitations in the methods section. In addition, some countries experienced the pandemic before others meaning their findings were more likely to be published at the time of this review. The knowledge of the impact of Covid-19 is growing daily, so this review in a way is an early insight which should be updated in the future.
Reviewer 3 Report
Dear authors,
this manuscript faces an interesting issue related to confinement effects, regarding stress, ansiety and depression, on population. You provide a friendly-reading manuscript with a detailled descrition of the Material and Methods.
However some facts could be taken into account to improve the text from a scientific sound point of view.
First, you provide a very short introduction related to state of the art. If you consider, you could provide more references related to this topic and the risk factors that you are considering.
Second, regarding to the considered risk factors, in my opinion, there are more interesting ones furthermore than those you are considering (female gender, young ages, no income and suffering psychiatric illness). For example, in a confinement due to this pandemic situation, variables related coexistence in the family nucleus, presence of children or pets, type of family home (small or not), access and use of ITC, close family member has died or is affected by COVID and others, are relevant risk factors for stress, ansiety or depression. I don't know if you are only considering these four risk factors because there is not enough literature related to the other risk factors or because a precise interest, please aclare.
Finally, once you perform a so detailled and well-done systematic review, since you are interested in ratios (prevalence) of ansiety, stress and depression for the different groups from the risk factors considered, in my opinion, you could perform a metaanalysis for these prevalences, providing a global estimation according to the groups in risk factors considered.
Reviewer 4 Report
Thank you for the hard work on the manuscript. However, I believe it requires some editing and clarification.

Round 2
Reviewer 2 Report
Thanks for the revised paper and for considering all the comments made, the paper looks much better for the revisions.
However, i remain concerned on a few points:
- The following terms should be changed throughout the paper:
a) risk factors Change To: factors that are associated with poor mental health (These studies are on the whole not causal they are exploratory).
b) protective factors; Change To: factors that are associated with good mental health
c) newly developed symptoms / increased mental ill health. Change To: symptoms of mental ill health. (We do not know if these symptoms are new or existed pre-covid).
d) Triggers; Change to: associations
2. The Discussion still over explains the findings. For example we know women may be more likely to have poor mental health, but this review has not looked for the reasons behind this, and whilst the suggestions in the discussion are reasonable, they are also hypothetical and include many assumptions from research unrelated to Covid-19. I would suggest deleting the discussion sections:
- Page 4 Line 298-315
- Page 5 Line 320-325
- Page 5 line 334-339
3. There are still some English language concerns to address throughout, which a careful proof read would help with, however the discussion does now read better.
Otherwise, congratulations on a very interesting and useful paper.
Author Response
"Please see the attachment"

Reviewer 3 Report
Dear authors,
you have improved your manuscript from a friendly-reading point of view.
First, I still think that there are not enough risk factors under study for the different levels of ansiety, so on. Perhaps, as another reviewer comments, if you focuse this study on sensible groups of population, it is more than sure that more risk factors will appear. On the other hand, I also still think that you could desing your review from a meta-analysis point of view because despite, as you mention in your answer, there is a lack of homogeinity for the different measure scales, this is not a problem since you have easy ways to get homogenity among them by means, for example, a normalization.
Finally, I think that you provide a well-written and well-done systematic review despite I also think that you could have faced a more ambicious challenge. May be for future.
This is the reason because, if the rest of reviewers and the editor support your work, I could recomend your manuscript for publication.
